# Thermodynamic Guardrails: Real-Time Monitoring of Physical Consistency in Stochastic Biochemical Models

**Gemini 2.5 Pro**
Google
1600 Amphitheatre Parkway
Mountain View, CA 94043, USA
`https://gemini.google.com`

**Liner AI**
Liner, Inc.
330 Townsend St.
San Francisco, CA94107, USA
`https://getliner.com`

**Jaeyoung Choi**
Korea Science Academy of KAIST
105-47 Baegyanggwanmun-ro
Busanjin-gu, Busan, Republic of Korea
`25-116@ksa.hs.kr`

## Abstract

The stochastic total quasi-steady-state approximation (stQSSA) is a vital model reduction technique for managing the computational complexity of multiscale biochemical networks. However, its validity is conditional, and it is known to fail in regimes of tight molecular binding and near-equimolar species concentrations Kim et al. [2014], Song et al. [2021]. This work introduces a thermodynamic guardrail, based on bond graph formalism, as a robust, first-principles-based diagnostic tool for detecting when the stQSSA produces a physically implausible state. Rooted in the Second Law of Thermodynamics, the guardrail identifies the stQSSA's failure by detecting negative power flow—the product of chemical affinity and flux—which signifies a thermodynamic violation. While the diagnosis is successful, the subsequent attempt to switch to a full Stochastic Simulation Algorithm (SSA) reveals a critical challenge: Even with the error detected, naive re-initialization is insufficient to correct the integrated error accumulated by the stQSSA, suggesting other, effective hand-off methods must be employed. This work validates thermodynamic monitoring as a crucial diagnostic for autonomous scientific agents, providing a "physical conscience" to prevent the propagation of erroneous results, while simultaneously exposing the non-trivial problem of state hand-off for the development of truly robust, self-correcting models.

## 1 Introduction

Kinetic modeling provides a foundational framework for systems biology, yet the scale of biochemical networks presents significant computational challenges. Consequently, model order reduction is essential for achieving computational tractability. A canonical technique is the stochastic total quasi-steady-state approximation (stQSSA), which is widely used to manage the computational burden of multiscale stochastic networks. However, the validity of the stQSSA is conditional. It has a well-characterized failure mode: when two species with similar total molecular counts bind together tightly, a large fraction of the unbound species becomes sequestered in the complex, and the algebraic nature of the stQSSA formula leads to a significant overestimation of the number of free molecules and an inaccurate prediction of system dynamics Kim et al. [2014], Song et al. [2021].

Typically, validating the stQSSA requires post-hoc comparison against a full stochastic simulation, identifying errors only after they have occurred. While advanced adaptive frameworks such as the

ASSISTER software package achieve universal validity, they rely on parameter-based thresholds to decide which approximation is valid in a given regime. Such switches require a priori knowledge of a model's validity conditions, which may not be available for novel, autonomously constructed models. This highlights the need for an intrinsic signal to guide adaptive model reduction that is grounded in fundamental physical principles rather than pre-analyzed conditions. The central hypothesis of this work is that a failing stQSSA model will produce system states that directly violate the Second Law of Thermodynamics, specifically by generating a negative power flow (the product of chemical affinity and flux, $A \cdot J < 0$). Detecting this thermodynamic violation can serve as a robust, physically principled trigger for model correction.

This paper therefore addresses the following research question: Can a model detect its own failure without prior knowledge of parameter-based validity regimes, by instead identifying a direct violation of a fundamental physical law? To investigate this, we develop a hybrid simulation model equipped with a thermodynamic guardrail and test it against the canonical substrate sequestration failure mode of the stQSSA. We demonstrate that the guardrail provides an immediate and definitive diagnosis of the approximation's failure. We then analyze the subsequent, flawed recovery, which reveals the profound challenge of state hand-off between approximate and full models. These findings establish the thermodynamic guardrail as a powerful diagnostic tool for building more physically robust and reliable autonomous modeling agents and clarify a key direction for future research in self-correcting systems.

## 2 Literature Review

### 2.1 Stochastic Model Reduction

Kinetic models of biochemical networks often contain reactions that occur on disparate timescales. For stochastic systems with low molecular copy numbers, direct simulation using the Gillespie algorithm is computationally demanding, as the vast majority of computational time is spent simulating fast, repetitive reactions (e.g., transcription factor binding and unbinding) to capture the dynamics of much slower processes of interest, such as transcription and translation. This "stiffness" necessitates model reduction techniques to achieve computational tractability Song et al. [2021], Kim et al. [2014].

The theoretically exact framework for this reduction is the slow-scale Stochastic Simulation Algorithm (ssSSA) Kim et al. [2014]. This approach eliminates fast reactions by reformulating the propensities of the slow reactions. The exact, or "ideal," stochastic quasi-steady-state approximation (QSSA) is derived by averaging the slow-reaction propensities over the conditional stationary probability distribution of the fast species, denoted $P(n_f|n_s)$. The resulting exact propensity is given by:

$$\bar{a}^s(n_s) = \sum_{n_f} a^s(n_f, n_s)P(n_f|n_s)$$

For reactions where the propensity function $a^s$ is linear in the fast species count $n_f$, this simplifies to replacing $n_f$ with its conditional expectation, $\langle n_f|n_s \rangle$. However, this ideal reduction is almost never practical, as the conditional distribution $P(n_f|n_s)$ and its moments are analytically intractable for all but the simplest systems. This intractability mandates the use of computationally feasible approximations Song et al. [2021].

### 2.2 Practical QSSA Methods

To circumvent the intractability of the exact stochastic QSSA, a widely adopted heuristic is to approximate the conditional average of fast species, $\langle n_f|n_s \rangle$, using algebraic formulas derived from deterministic QSSAs. This approach gives rise to a hierarchy of approximations with varying degrees of accuracy and specific ranges of validity. The most prominent of these are the stochastic Standard QSSA (sQSSA), the stochastic Total QSSA (stQSSA), and the stochastic Low-state QSSA (sIQSSA).

The stochastic Standard QSSA (sQSSA) was an early approach that used propensities derived from the classic Michaelis-Menten rate law. This method is now known to be the least accurate, as its underlying deterministic formulation is only valid under restrictive conditions, such as negligible enzyme concentration, and fails to properly account for the sequestration of species in complexes Song et al. [2021]. Its propensity functions are highly sensitive to fluctuations in slow species counts, often leading to significant errors Kim et al. [2014].

The stochastic Total QSSA (stQSSA) became the preferred method due to its enhanced accuracy. It uses a more robust algebraic formula derived from the deterministic total QSSA, which correctly accounts for the conservation of total molecular counts (e.g., $A_T = A_{free} + C_{complex}$). Below is the stQSSA for $[ES]$ in irreversible Michaelis-Menten kinetics:

$$[ES] \approx [ES]_{tqssa} = \frac{1}{2} \left\{ ([S]_t + [E]_t + K_M) - \sqrt{([S]_t + [E]_t + K_M)^2 - 4[S]_t[E]_t} \right\}$$

Where $K_M = (k_{-1} + k_2)/k_1$ is the Michaelis constant and $[S]_t, [E]_t$ are the total concentrations of substrate and enzyme, respectively. The propensity, or rate, of the equation is calculated as $v_{tqssa} = k_2[ES]_{tqssa}$. Additionally, free substrate $[S]_{free}$ can be found using $[S]_t - [ES]$.

This method is significantly less sensitive to molecular fluctuations than that of the sQSSA, providing an accurate approximation for a much broader range of conditions Kim et al. [2014]. This robustness led to the widespread belief that the stQSSA was a universally valid approximation for stochastic models Song et al. [2025].

However, recent work has definitively shown that the stQSSA is not universally valid. It has a specific, well-characterized failure mode: when two species with similar total molecular counts ($A_T \approx B_T$) bind together tightly (i.e., the dissociation constant $K_d$ is small). Quantitatively, the stQSSA becomes inaccurate when the dimensionless parameter $A_T K_d$ is less than approximately 10 Song et al. [2021]. In this regime, a large fraction of the unbound species is sequestered in the complex, and the algebraic nature of the stQSSA formula leads to a significant overestimation of the number of free molecules Song et al. [2025]. To address this specific failure, the stochastic Low-state QSSA (sIQSSA) was developed. The sIQSSA is founded on the physical observation that in the tight-binding regime, the unbound species spends almost all of its time at a copy number of zero or one Song et al. [2021]. This "low-state" assumption yields a different algebraic approximation that is highly accurate precisely when the stQSSA fails. For a reversible binding reaction A + B $\rightleftharpoons$ C, the two-state sIQSSA approximation for the average count of species A, $\langle A \rangle_{lq}$, is given by:

$$\langle A \rangle_{lq} = \begin{cases} \frac{(A_T - B_T + 1)(A_T - B_T + B_T K_d)}{A_T - B_T + A_T K_d + 1} & \text{if } A_T \geq B_T \\ \frac{A_T K_d}{B_T - A_T + A_T K_d + 1} & \text{if } A_T < B_T \end{cases} \tag{1}$$

This has led to state-of-the-art adaptive frameworks, such as the ASSISTER software package, which achieves universal validity by automatically selecting the stQSSA when $A_T K_d > 10$ and switching to the more accurate sIQSSA when $A_T K_d < 10$ Song et al. [2021].

## 2.3 Self-Correcting Models

The concept of a model that can assess its own validity and adapt its strategy mid-simulation is a well-established paradigm in computational science. This approach is analogous to adaptive step-size solvers for ordinary differential equations, which automatically adjust their temporal resolution to maintain a desired level of accuracy. It is also conceptually similar to adaptive mesh refinement in physics simulations, where computational resources are dynamically focused on regions of high activity. In mathematical biology, this has largely taken the form of frameworks that switch between different pre-defined approximations, as seen with ASSISTER.Song et al. [2021] These models rely on parameter-based thresholds to decide which approximation is valid in a given regime. The approach explored in this work represents a different direction, focusing on self-diagnosis based on fundamental physical law rather than pre-analyzed parameter regimes. Instead of relying on parameter-based validity conditions, detection of failures by identifying direct violations of a fundamental physical law—in this case, the Second Law of Thermodynamics-is proposed. This establishes a proof-of-concept for a model-agnostic trigger that is grounded in first principles rather than pre-analyzed conditions.

## 2.4 Thermodynamics and Bond Graphs as a Foundation for Model Validity

The inherent difficulties in modeling complex biological systems have led to a call for frameworks that are grounded in more fundamental, physical principles rather than relying solely on heuristic or phenomenological descriptions.Vittadello and Stumpf [2022] Because living systems are fundamentally dissipative—they constantly consume and turn over energy—the laws of thermodynamics impose tight and non-negotiable constraints on the behavior of any valid biological model.

A powerful and explicit framework for enforcing these constraints is the bond graph formalism. As highlighted by Vittadello and Stumpf in their survey of open challenges in the field, bond graphs offer a "flexible framework for thermodynamically correct models" by explicitly tracking energy and mass flow.Vittadello and Stumpf [2022] This approach has several key advantages: it ensures that resulting models incorporate stoichiometry and conservation of energy from the outset, and it leads to models that are inherently modular, mirroring the structure of real biological systems. This provides a rigorous and well-defined pathway for developing biophysically and thermodynamically consistent models of molecular processes. While the foundational work on network thermodynamics and bond graphs was established decades ago Paynter [1961], Oster et al. [1973], its application to the challenges of modern systems biology represents a key area for future research.

While bond graphs have been primarily discussed as a tool for building thermodynamically consistent models from the ground up, their potential for the real-time, dynamic validation of approximate models remains a novel area of exploration. This work addresses this specific gap, proposing that the thermodynamic principles inherent in the bond graph formalism can be repurposed to serve as an internal consistency check—a "guardrail"—to detect when a computationally efficient but simplified model has entered a physically implausible state.

## 3 Methods

To test the efficacy of a thermodynamic guardrail, three distinct simulation models were developed and compared: a physically accurate stochastic ground truth model, a computationally efficient approximate model, and a novel hybrid model designed to diagnose physical inconsistency and trigger a switch to the full simulation.

### 3.1 The Stochastic Ground Truth Model

The benchmark for physical accuracy is a discrete stochastic simulation of the complete Michaelis-Menten reaction network:

$$E + S \underset{k_{-1}}{\overset{k_1}{\rightleftharpoons}} ES \underset{k_{-2}}{\overset{k_2}{\rightleftharpoons}} E + P \tag{2}$$

This model was implemented using the Gillespie Stochastic Simulation Algorithm (SSA).Gillespie [1977] The SSA provides a statistically exact numerical realization of the system's chemical master equation. All comparisons are made against an ensemble of 450 independent SSA trajectories, which is defined as the ground truth.

### 3.2 The Approximate Model: stQSSA

The computationally efficient model was implemented using the stochastic total quasi-steady-state approximation (stQSSA). The propensities for product formation was calculated as $v_{tqssa} = k_2[ES]_{tqssa}$ (forward) and $k_{-2}[E][P]$ (reverse), where $[ES]_{tqssa}$ is the quasi-steady-state concentration of the enzyme-substrate complex derived from the total species concentrations and $[E]$ is derived from it using total species conservation.

### 3.3 The Self-Correcting Hybrid Model

The hybrid model begins simulation using the stQSSA and performs a physical consistency check at each time step. The procedure is as follows:

**1. State Reconstruction:** The full system state ($[E], [S], [ES], [P]$) is reconstructed from the reduced state using the tQSSA solution and mass conservation laws, as detailed in Appendix A.

**2. Thermodynamic Evaluation:** The chemical affinity ($A_j$) is calculated for the two underlying elementary reactions $E + S \rightleftharpoons ES$ (binding) and $ES \rightleftharpoons E + P$ (catalysis).

**3. Consistency Check:** The thermodynamic power, $P_j = A_j \cdot J_j$, is evaluated post-reaction for each elementary reaction and the net reaction. If either $P_{binding} < 0$ or $P_{catalysis} < 0$ or $P_{net} < 0$, a thermodynamic violation is triggered. The type of guardrail was set to 'net' in this work, meaning that only the net power violation was used as the trigger. Upon trigger, the simulation immediately and permanently switches from the stQSSA solver to the full Gillespie SSA. Crucially, the hybrid

model discards the physically implausible reconstructed state and re-initializes the full SSA from first principles using only the conserved total quantities. Because the reduced stQSSA model exclusively evolves the integer count of the product $[P(t)]$, and the total enzyme $[E_T]$ and initial substrate $[S_0]$ are integer parameters, the re-initialization state $([E_T, S_0 - P(t), 0, P(t)])$ is guaranteed to be integer-valued. This specific implementation detail elegantly sidesteps the general problem of continuous-to-integer state mapping, allowing for a direct and mass-conserving hand-off to the discrete-state SSA.

The computational efficiency was quantified by computing the mean wall-clock time per simulation over an ensemble of 450 simulations for the full SSA, pure stQSSA, and hybrid models.

### 3.4 Experimental Parameters

To create a stringent test for the guardrail, simulations were designed to induce an initial-state failure of the stQSSA. The following initial conditions and kinetic parameters were selected to create a tight-binding, equimolar regime where the stQSSA is known to be inaccurate Song et al. [2021]:

- Initial substrate concentration: $[S_0] = 10$
- Total enzyme concentration: $[E_T] = 10$
- Kinetic rates: $k_1 = 100$, $k_{-1} = 1$, $k_2 = 1$, $k_{-2} = 0.01$

The equimolar concentrations ($[E_T] = [S_0]$) and low dissociation constant ($K_d = k_{-1}/k1 = 0.01$) ensure that from the initial time step, a substantial fraction of free species is sequestered into the ES complex. This provides an immediate and severe test of the guardrail's diagnostic capability.

### 3.5 Parameter Sweeps

To assess the robustness of each guardrail type against many parameter regimes, parameter sweeps across three reaction rates were performed. $k_1, k_2, k_{-1}$ were sampled from log-uniform distributions over the ranges $[10^{-2}, 10^2]$ for $k_1, k_2$, and $k_{-1}$. For each heatmap, 2 parameters out of the three were varied, while the third was fixed to the median value of its range. Fixing $K_{eq}$ to 10000 (The value of $K_{eq}$ in this study's demonstration), $k_{-2}$ was derived directly using the Haldane relationship. Once all 4 parameters were fixed, 10 independent simulations were executed for each of the three models (SSA, stQSSA, Hybrid) and the IAE (Integrated Average Error) in product concentration at the final time point ($t = 50$) relative to the SSA was computed.

## 4 Results

### 4.1 Error of the stQSSA and Hybrid Models

A comparative analysis of the species concentration trajectories reveals the canonical failure mode of the stQSSA under the chosen tight-binding, equimolar conditions. The ground truth Stochastic Simulation Algorithm (SSA) correctly demonstrates the physical phenomenon of substrate sequestration; upon initiation, the concentration of the enzyme-substrate complex [ES] rises sharply as the concentrations of free enzyme [E] and free substrate [S] are rapidly depleted (Figure 2, black lines).

In contrast, the stQSSA (red in 2) fails to capture this sequestration. It underestimates the mean concentration of the $[ES]$ complex while consequently overestimating the concentration of the final product $[P]$. The fast species $[E]$ and $[S]$ are also incorrectly predicted to react slower than in the SSA. The stQSSA underestimates the formation of the $[ES]$ complex and, as a result, overestimates the amount of free enzyme $[E]$ available in the system.

### 4.2 Error Diagnosis via Thermodynamic Guardrail

The thermodynamic guardrail reliably identifies the physical inconsistency of the stQSSA's reconstructed state. As shown in 1, the net power (green dash-dotted line), which must remain non-negative according to the Second Law of Thermodynamics, drops below zero at $t \approx 2.7$ seconds. However, the early errors in predicting $[S], [E]$ and $[ES]$ concentrations are not flagged as thermodynamically inconsistent. This demonstrates that the guardrail is sensitive to violations of the Second Law of

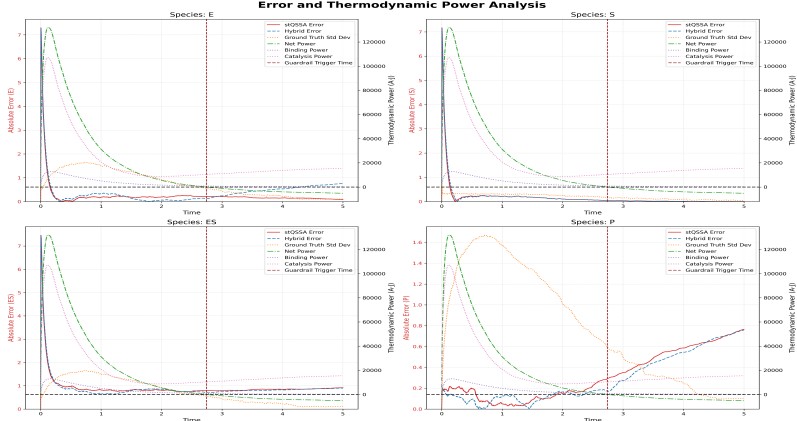

Figure 1: Quantitative analysis of model performance across all species. Each subplot shows the absolute error for the pure stQSSA (red) and hybrid (blue) models on the left axis. The right axis displays the thermodynamic powers for the stQSSA simulation, including net power (green), which serves as the physical consistency metric. The trigger time (vertical line) indicates the first detected violation.

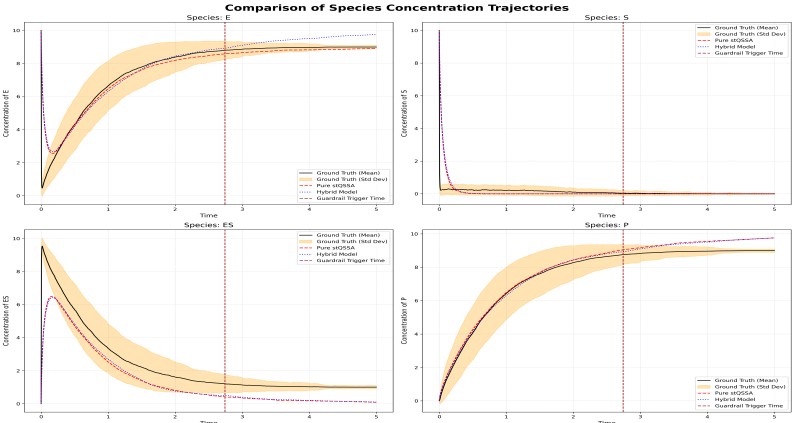

Figure 2: Comparison of product concentration trajectories. The mean trajectory of the stochastic ground truth is shown in black, with its standard deviation as a shaded orange region. The pure stQSSA trajectory (red) and the hybrid model trajectory (blue) are overlaid, demonstrating the reactive nature of the hybrid model. The trigger time (vertical line) indicates when the hybrid model switches to the full SSA.

Thermodynamics, not necessarily to all forms of numerical or kinetic error. The stQSSA's initial state, while kinetically inaccurate, is not yet thermodynamically impossible.

### 4.3 Post-Diagnosis Behavior

While the diagnosis of model failure is successful, the subsequent attempt at course correction is flawed. Following the guardrail trigger at $t \approx 2.7s$, the hybrid model switches to the full SSA solver, re-initializing the system using the conserved total quantities $[E_T]$ and $S_T = S_0 - P(t)$. However, this does not correct the trajectory onto the SSA-it rather veers off into a new physically consistent trajectory that diverges from the ground truth, especially prominent in $[E]$ trajectories.

### 4.4 Computational Performance and Parameter Sensitivity

In terms of computational efficiency, both the stQSSA and the hybrid models demonstrate a clear advantage over the full SSA. The mean wall-clock time per simulation was 0.0005s for the SSA, compared to nearly identical runtimes of 0.0003s for both the pure stQSSA and the hybrid model.

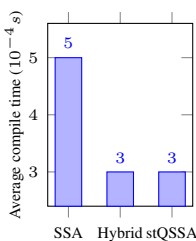

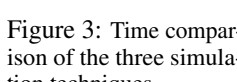

Figure 3: Time comparison of the three simulation techniques.

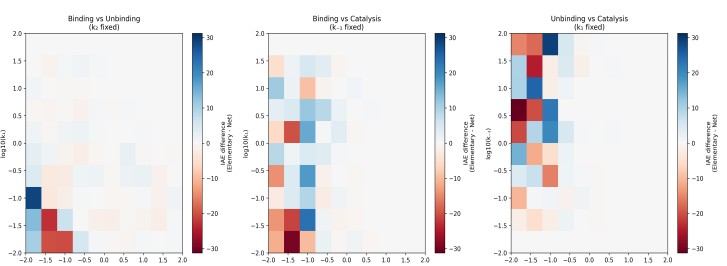

Figure 4: Heat map of IAE between stQSSA and net reaction guardrail. For each parameter combination, 10 runs occurred with all 3 models. the non-swept parameters were held constant at median. $K_{eq}$ was set to $10^4$ for all cases.

This confirms that in this simulation regime, the overhead of performing the thermodynamic check at each step is negligible.

The parameter sweep analysis (Figure 4), which compares the performance of an 'elementary' versus a 'net' power guardrail, yields ambiguous results. The heatmaps show a mixture of blue regions (where the net guardrail performs better) and red regions (where it performs worse).

The complete, anonymized source code and instructions to reproduce all simulations and figures are provided in the supplementary material.

## 5 Discussions

### 5.1 The Role of Thermodynamic Consistency as a Diagnostic

When the system was initialized under conditions of tight binding and equimolar concentrations—a regime where the stQSSA is known to fail due to substrate sequestration—the guardrail detected a violation of the Second Law of Thermodynamics at approximately $t \approx 2.7s$. At the moment of this trigger, the concentration trajectories of all four species in the hybrid model are still within or near the one-standard-deviation bounds of the ground truth SSA ensemble (see 2). This suggests the guardrail can detect physical inconsistency before the model's trajectory deviates significantly from true stochastic dynamics. While the guardrail did not detect the initial kinetic errors in the fast species, it successfully triggered before the error in the slow variable, the product $[P]$, became catastrophic. This positions it as a crucial check for preventing the long-term propagation of physical impossibilities, even if it is not sensitive to all forms of short-term numerical error.

This finding's significance lies in the guardrail's model-agnostic nature. Unlike parameter-based switches that require a priori knowledge of a model's specific validity conditions (e.g., the $A_T \cdot K_d < 10$ rule for ASSISTER), the thermodynamic check is universal. It functions as a "physical conscience" for a simulation, assessing its validity against a fundamental law rather than a pre-analyzed parameter map. This provides an intrinsic, online certification of physical plausibility, a critical capability for autonomous agents tasked with building and simplifying novel kinetic models whose failure regimes are not yet known.

### 5.2 Analysis of Correction Failure

While the guardrail's diagnosis was successful, the subsequent attempt at correction led to a new challenge in hybrid modeling: the integrated error problem. Interestingly, at the time of violation, only $[ES]$ is significantly erroneous. It is hypothesized that while the hybrid model predicts one more molecule of $[E]$ to make up for this error, this does not fully account for the overall integrated error, demonstrating the need for a more sophisticated hand-off mechanism.

The full SSA is initialized onto a new, physically self-consistent trajectory, one that corresponds to a different physical reality—a system with a lower initial amount of $[ES]$ (see 2). This trajectory causes divergence in $[E]$, effectively increasing the error. Alternative correction mechanisms must be employed to successfully bridge the gap from diagnosis to true self-correction.

### 5.3 Guardrail Sensitivity

Given that the correction mechanism is flawed, Figure 4 does not point to a conclusively superior guardrail type. Rather, the heatmap illustrates the consequences of the flawed correction mechanism. During the parameter sweep, the 'elementary' guardrail was never observed to be triggered. Thus, the red regions, where the net guardrail performs worse, likely represent parameter regimes where the error introduced by the flawed state hand-off exceeds the error of the uncorrected stQSSA.

The fact that the elementary guardrail never triggered suggests the stQSSA's algebraic formulation, while inaccurate, retains internal consistency for the individual reaction steps. It also implies that the heatmap is actually measuring the sensitivity of the net guardrail to the integrated error problem, rather than its fundamental diagnostic capability.

### 5.4 Proposal for Accurate Hand-Offs in Self-Correcting Models

The analysis shows that a successful hand-off must contend with the integrated error, particularly in $[ES]$, rendering a simple switch insufficient. More sophisticated algorithms are needed. Furthermore, a proactive approach could identify predictive mathematical features within the thermodynamic properties themselves. For instance, the local minimum observed in the catalysis power graph around $t \approx 1.5s$ could, with rigorous validation, serve as an earlier, more predictive trigger than a direct thermodynamic violation. Another proposed strategy is "pause-and-burn-in": upon a trigger, the slow variable $[P]$ is halted while the SSA uses only fast reversible reactions ($E + S \rightleftharpoons ES$) to find the correct sequestered equilibrium. The slow reactions are then unpaused, allowing the simulation to continue from this correctly equilibrated state with the full SSA.

### 5.5 Significance for Autonomous Scientific Agents

These findings have profound implications for the development of autonomous scientific agents. By equipping an AI agent with thermodynamic consistency monitoring, it obtains a mechanism to self-validate its models against first principles. An agent building a novel biochemical network will not know its failure regimes in advance. The guardrail acts as a universal "check engine light," flagging an unknown problem based on a definitive violation of physical law. This ensures the agent does not unknowingly propagate physically impossible results, a critical enabler for reliable, large-scale autonomous model construction. There are also potential dangers in this system, however. In constructing the 'guardrail', the AI agent often hallucinated equalities or inequalities. Such problems may suggest that while the guardrail provides a powerful diagnostic, it must be implemented with careful oversight to avoid misinterpretation or over-reliance on potentially flawed AI-generated logic.

## 6  Conclusion

This study introduces and validates a thermodynamic guardrail as a robust, first-principles-based diagnostic for failures in stochastic biochemical models. By embedding a consistency check rooted in the Second Law of Thermodynamics, an adaptive hybrid model successfully and reliably detects a canonical failure of the stQSSA arising from substrate sequestration. This approach provides a model-agnostic check that operates without prior knowledge of system-specific failure regimes, offering a distinct advantage over parameter-based methods.

These findings have significant implications for the development of autonomous scientific agents. Equipping such agents with a "physical conscience" ensures that their explorations remain anchored in fundamental physical law, preventing the propagation of erroneous results from failed approximations. While advanced approximations like the sIQSSA are powerful in known failure regimes, the thermodynamic guardrail offers a universal, complementary validation layer. Future work must now focus on developing robust state hand-off algorithms to bridge the gap from successful diagnosis to true self-correction, advancing the creation of the next generation of reliable, autonomous scientific discovery systems.

Two AI models were used in the research process: Liner AI and Google Gemini. For Google Gemini, a custom Gem, with specifications that enhanced its credibility, was used. Specifically, Liner's Hypothesis Generator, Literature Review, and Peer Review features were used to iteratively refine the initial hypothesis, search for citations to structure the Introduction and Literature Review sections of

the paper, and review and refine the paper after initial draft completion respectively. The Gemini tool assisted mainly in generating code, verifying mathematical rigor, and generating the text used for most of the paper.

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

# A Theoretical Framework of the Thermodynamic Guardrail

This appendix provides the detailed theoretical and methodological underpinnings of the thermodynamic guardrail. First, a bond graph representation of the Michaelis-Menten reaction network is presented, illustrating how chemical potentials, affinities, and fluxes are defined within this formalism. Next, the derivation of the thermodynamic power metrics used for the guardrail check is outlined. The core consistency check is first derived from fundamental thermodynamic principles. Finally, an algorithmic summary of the guardrail implementation is provided.

## A.1 Bond Graph Formalism for Biochemical Systems

**Overview of Bond Graphs.** Bond graphs are a domain-independent language for modeling physical systems that explicitly tracks energy flow Paynter [1961], Oster et al. [1973]. The elements provide a direct graphical representation of the thermodynamic concepts defined above.

- **C (Capacitor):** Represents a chemical species storing energy, defined by its chemical potential, $\mu_i$.
- **Re (Resistor):** Represents a reaction dissipating energy, defined by the relationship between its affinity ($\mathcal{A}_j$) and flux ($J_j$).
- **0- and 1-Junctions:** These represent Kirchhoff's laws for chemical systems, enforcing conservation of mass and defining how chemical potentials combine to create reaction affinities.

**Full Elementary Model.** The full elementary network is represented by the bond graph in Figure 5. The affinities for the two elementary reactions, derived from the 1-junctions, are:

$$\mathcal{A}_1 = \mu_E + \mu_S - \mu_{ES} \tag{3}$$
$$\mathcal{A}_2 = \mu_{ES} - (\mu_E + \mu_P) \tag{4}$$

The corresponding fluxes for these reactions are:

$$J_1 = k_1[E][S] - k_{-1}[ES] \tag{5}$$
$$J_2 = k_2[ES] - k_{-2}[E][P] \tag{6}$$

The guardrail check involves calculating $P_1 = \mathcal{A}_1 \cdot J_1$ and $P_2 = \mathcal{A}_2 \cdot J_2$ to ensure both are non-negative.

**Reduced stQSSA Model.** The reduced 'stQSSA' model (Figure 6) lumps the entire process into a single dissipative element, $Re_{MM}$. While this is computationally efficient, it obscures the underlying elementary steps. The guardrail works by reconstructing the state of these hidden steps to check their physical validity.

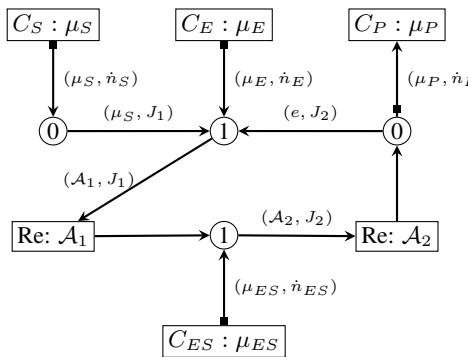

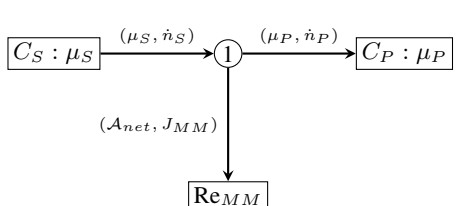

Figure 6: Bond graph of the reduced stQSSA model $S \rightleftharpoons P$.

Figure 5: Bond graph of the full elementary reaction network $E + S \rightleftharpoons ES \rightleftharpoons E + P$.

## A.2 Thermodynamic Validation of Parameters and Power

The guardrail is a direct application of the Second Law of Thermodynamics to the kinetics of a biochemical system. The derivation proceeds by linking the thermodynamic driving force of a reaction to its kinetic rate. First, the Haldane relationship between parameters connects the equilibrium constant of a reaction to its kinetic rate constants, ensuring thermodynamic consistency at equilibrium. When the Haldane relationship is satisfied, the thermodynamic affinity and flux for each elementary reaction can be determined, leading to the definition of thermodynamic power as their product. The Second Law requires that this power be non-negative for all physically plausible states.

**Chemical Potential and Reaction Affinity.** The thermodynamic state of the system is defined by the chemical potential, $\mu_i$, of each species $i$, given by:

$$\mu_i = \mu_i^0 + RT \ln[i] \tag{7}$$

where $\mu_i^0$ is the standard chemical potential, $R$ is the ideal gas constant, $T$ is the temperature, and $[i]$ is the species concentration. To ensure numerical stability when calculating chemical potentials, a small regularization term, $\varepsilon$, is used for species with zero counts. The concentration term in 7 is therefore implemented as $\max([i], \varepsilon)$, where $\varepsilon$ was set to $1.0 \times 10^{-9}$ in the simulations. This standard procedure prevents undefined logarithmic calculations while having a negligible impact on the thermodynamic forces of species with non-zero counts. The net thermodynamic driving force for any elementary reaction is its affinity, $\mathcal{A}$, defined as the negative of the Gibbs free energy change of the reaction ($\mathcal{A} = -\Delta_r G$). For the forward binding reaction $E + S \to ES$, the affinity $\mathcal{A}_1$ is:

$$\mathcal{A}_1 = (\mu_E + \mu_S) - \mu_{ES} \tag{8}$$

A positive affinity ($\mathcal{A}_1 > 0$) indicates that the forward reaction is thermodynamically favored.

**Haldane Relationship.** In a reversible reaction, the equilibrium constant, $K_{eq}$, is related to the forward and reverse kinetic rate constants, because the net flux is zero at equilibrium. Thus, in the general reaction $A + B \underset{k_r}{\overset{k_f}{\rightleftharpoons}} C$,

$$K_{eq} = \frac{k_f}{k_r} = \frac{[C]_{eq}}{[A]_{eq}[B]_{eq}} \tag{9}$$

Additionally, in thermodynamic equilibrium, the affinity for the reaction is zero ($\mathcal{A} = 0$), indicating no net driving force. Thus, $\mathcal{A} = \mu_A + \mu_B - \mu_C = 0$ at equilibrium. Substituting $\mu_i$ from Equation 7 allows for derivation of the following:

$$\mu_A^0 + \mu_B^0 - \mu_C^0 = RT \ln \left( \frac{[C]_{eq}}{[A]_{eq}[B]_{eq}} \right) = RT \ln(K_{eq}) \tag{10}$$

Substitute the definition of Gibbs free energy change at standard conditions, $\Delta_r G^0 = \mu_C^0 - (\mu_A^0 + \mu_B^0)$, into Equation 10 yields the well-known relationship:

$$\Delta_r G^0 = -RT \ln(K_{eq}) \tag{11}$$

This relationship links the standard Gibbs free energy change of a reaction to its equilibrium constant, ensuring thermodynamic consistency.

For the binding reaction $E + S \underset{k_{-1}}{\overset{k_1}{\rightleftharpoons}} ES$, the Haldane relationship is given by:

$$\frac{k_1}{k_{-1}} = K_{eq,1} \implies \Delta_r G_1^0 = \mu_{ES}^0 - \mu_E^0 - \mu_S^0 = -RT \ln \left( \frac{k_1}{k_{-1}} \right) \tag{12}$$

Similarly, for the catalytic reaction $ES \underset{k_{-2}}{\overset{k_2}{\rightleftharpoons}} E + P$, the relationship is:

$$\frac{k_2}{k_{-2}} = K_{eq,2} \implies \Delta_r G_2^0 = \mu_E^0 + \mu_P^0 - \mu_{ES}^0 = -RT \ln \left( \frac{k_2}{k_{-2}} \right) \tag{13}$$

Satisfying these relationships ensures that the model parameters are thermodynamically consistent at equilibrium. For the model tested in this work, Gibbs number $g_i = \frac{\mu_i^0}{RT}$ is defined for ease of calculation, with $g_S = g_E = \mu_S^0 = \mu_E^0 = 0$. Also, $K_{eq,1} = k_1/k_{-1} = 100$ and $K_{eq,2} = k_2/k_{-2} = 100$. This allows calculation of $\mu_{ES}^0 = -RT \ln 100$ and $\mu_P^0 = -2RT \ln(100)$. In the code, $g_{ES}$ and $g_P$ are calculated as follows using the above relationships.

$$g_{ES} = -\ln(k_1/k_{-1}) \, (= -\ln(100)) \tag{14}$$
$$g_P = -\ln(k_2/k_{-2}) + g_{ES} \, (= -2\ln(100)) \tag{15}$$

**Physical Consistency.** The crucial link between thermodynamics and kinetics is the Thermodynamic Power, $P$, which is the product of the thermodynamic force (Affinity) and the resulting flow (Flux):

$$P_j = \mathcal{A}_j \cdot J_j \tag{16}$$

The Second Law of Thermodynamics requires that the net flux of a spontaneous process must occur in the same direction as the thermodynamic driving force. This means that if the affinity $\mathcal{A}_j$ is positive, the flux $J_j$ must also be positive (or zero at equilibrium), and vice versa. In all cases, their product, the thermodynamic power, must

be non-negative. This establishes the fundamental consistency condition that any physically plausible model must obey:

$$P_j = \mathcal{A}_j \cdot J_j \geq 0 \tag{17}$$

Given the Haldane relationship is satisfied, a simulation state that produces a negative power ($P_j < 0$) is therefore describing a physically impossible event: a net flow of molecules against their own thermodynamic potential gradient. This violation serves as a definitive, non-arbitrary signal that the model approximation has failed.

## A.3 Derivation of the stQSSA and Implementation of the Guardrail

**The stQSSA.** The stQSSA is an approximation that assumes that binding/unbinding reactions are much faster than product formation. Under this assumption, the concentration of the enzyme-substrate complex $[ES]$ rapidly reaches a quasi-steady state relative to the slower dynamics of substrate $[S]$ and product $[P]$. The algebraic expression for $[ES]$ in the stQSSA is thus derived by setting the time derivative of $[ES]$ to zero and solving for $[ES]$:

$$\frac{d[ES]}{dt} = k_1[E][S] - k_{-1}[ES] - k_2[ES] + k_{-2}[E][P] \approx 0 \tag{18}$$

Here, the total enzyme and substrate conservation, $[E_T] = [E] + [ES]$ and $[S_T] = [S] + [ES]$, are used to formulate a quadratic equation in $[ES]$:

$$(k_1([S_T] - [ES]) + k_{-2}[P])([E_T] - [ES]) - (k_{-1} + k_2)[ES] = 0 \tag{19}$$

$$[ES]^2 - ([E_T] + [S_T] + K_M + \frac{k_{-2}}{k_1}[P])[ES] + [E_T]([S_T] + \frac{k_{-2}}{k_1}[P]) = 0 \tag{20}$$

where $K_M = \frac{k_{-1}+k_2}{k_1}$ is the Michaelis constant. Solving this quadratic equation yields the stQSSA expression for $[ES]$:

$$[ES] = \frac{([E_T] + [S_T] + K_M + \frac{k_{-2}}{k_1}[P]) - \sqrt{([E_T] + [S_T] + K_M + \frac{k_{-2}}{k_1}[P])^2 - 4[E_T]([S_T] + \frac{k_{-2}}{k_1}[P])}}{2} \tag{21}$$

With $[ES]$ expressed algebraically, the effective (net) stQSSA propensity $v_{stQSSA}$ for product formation is given by $k_2[ES] - k_{-2}[E][P]$, allowing the reduced model to simulate only the dynamics of $[S]$ and $[P]$. To check the 'stQSSA''s validity at any given time step, the full system state from the reduced state is algorithmically reconstructed and tested for violations of the Second Law (Eq. 17).

**State Reconstruction.** At time $t$ during the simulation, there are two knowns, $[E_T]$ and $[P(t)]$ from which the full state is reconstructed. State reconstruction begins with calculating the slow variable $[S_T] = [S_0] - [P(t)]$. Then, using the stQSSA algebraic expression, $[ES(t)]$ and subsequently $[E(t)], [S(t)]$ is computed:

$$\frac{([E_T] + [S_T] + K_M + \frac{k_{-2}}{k_1}[P]) - \sqrt{([E_T] + [S_T] + K_M + \frac{k_{-2}}{k_1}[P])^2 - 4[E_T]([S_T] + \frac{k_{-2}}{k_1}[P])}}{2} \tag{22}$$

$$[E(t)] = [E_T] - [ES(t)] \tag{23}$$

$$[S(t)] = [S_T] - [ES(t)] \tag{24}$$

This step bridges the gap between the reduced model's state and the full system's physical description, testing the internal consistency of the approximation's own logic.

**Thermodynamic Evaluation and Consistency Check.** Reconstructing the complete state vector ($[E], [S], [ES], [P]$), the chemical potential $\mu_i$ for each species is calculated using Eq. 7. From these, the affinities $\mathcal{A}_1$ and $\mathcal{A}_2$ are computed for the two underlying elementary reactions. Simultaneously the net fluxes $J_1$ and $J_2$ are determined based on the reconstructed concentrations. Finally, the thermodynamic power is computed for each elementary reaction and the net reaction:

$$P_1(t) = \mathcal{A}_1(t) \cdot J_1(t) \tag{25}$$

$$P_2(t) = \mathcal{A}_2(t) \cdot J_2(t) \tag{26}$$

$$P_{net}(t) = P_1(t) + P_2(t) \tag{27}$$

If either $P_1(t) < 0$ or $P_2(t) < 0 \, or \, P_{net} < 0$, the state is deemed physically inconsistent. This violation of the Second Law of Thermodynamics serves as the definitive trigger to switch from the fast 'stQSSA' solver to the full, physically accurate SSA solver. This work also separately tested the elementary reaction triggers ($P_1 < 0$ or $P_2 < 0$) versus the net reaction trigger ($P_{net} < 0$) to evaluate their relative effectiveness.

# Agents4Science AI Involvement Checklist

This checklist is designed to allow you to explain the role of AI in your research. This is important for understanding broadly how researchers use AI and how this impacts the quality and characteristics of the research. **Do not remove the checklist! Papers not including the checklist will be desk rejected.**

1. **Hypothesis development**: Answer: [C]

   Explanation: The human author initiated the process by providing the AI with a review paper on open problems in mathematical biology. Vittadello and Stumpf [2022] The AI was tasked with identifying promising research avenues within that context. It proposed several topics, from which the human selected 'bond graphs' for further exploration. A second AI agent was then prompted to generate specific, novel hypotheses at the intersection of bond graphs and model reduction, from which the final hypothesis of the paper was selected. The AI's role was therefore central to the conceptual synthesis and ideation.

2. **Experimental design and implementation**: Answer: [C]

   Explanation: The experimental framework, comparing a ground truth model against an approximate model and a self-correcting hybrid, was designed by the human co-author. The AI agent then implemented this design, writing the complete simulation code for all three models in Python. While the human set the initial parameters, the AI that analyzed them and correctly identified that they would induce the desired initial-state failure due to substrate sequestration, thus validating the experimental design.

3. **Analysis of data and interpretation of results**: Answer: [C]

   Explanation: The human co-author defined the high-level analysis plan, specifying the key metrics to be plotted (e.g., absolute error). The AI agent executed this plan by writing all necessary code to process the raw simulation data, perform the error calculations, and generate the final plots. The initial textual interpretation of these plots was also generated by the AI, but required significant correction by the human co-author due to the AI's limitations in accurately interpreting visual data.

4. **Writing**: Answer: [D]

   Explanation: The text of this paper was generated by an AI agent based on a structured outline, the generated figures, and a summary of their interpretation. Human involvement was limited to iterative prompting and copy-editing for flow, scientific accuracy, and the removal of AI-generated artifacts or 'hallucinations'.

5. **Observed AI Limitations**: Description: The AI agent demonstrated several key limitations that required human intervention. 1) Lack of Multimodal Reasoning: The AI was unable to accurately analyze the visual content of plots or the provided bond graph diagrams, often hallucinating features that were not present. 2) Deficient Abstract Reasoning: The agent, despite proposing ideas, could not follow through novel mathematical derivations. 3) Poor Iterative Design: When tasked with correcting its own errors in interpretation or writing, the agent often repeated the same mistakes. 4) Scalability of Task Execution: The AI performed best on narrowly-defined tasks. Complex, multi-stage goals had to be broken down into a sequence of smaller prompts by the human director.

